# Temperature Sensitivity and Composition of Nitrate-Reducing Microbiomes from a Full-Scale Woodchip Bioreactor Treating Agricultural Drainage Water

**DOI:** 10.3390/microorganisms9061331

**Published:** 2021-06-18

**Authors:** Arnaud Jéglot, Sebastian Reinhold Sørensen, Kirk M. Schnorr, Finn Plauborg, Lars Elsgaard

**Affiliations:** 1Department of Agroecology, Aarhus University, Blichers Allé 20, 8830 Tjele, Denmark; finn.plauborg@agro.au.dk (F.P.); lars.elsgaard@agro.au.dk (L.E.); 2WATEC Centre for Water Technology, Department of Agroecology, Aarhus University, Blichers Allé 20, 8830 Tjele, Denmark; 3Novozymes A/S, Biologiens Vej 2, 2800 Kongens Lyngby, Denmark; srsq@novozymes.com (S.R.S.); KkSc@novozymes.com (K.M.S.)

**Keywords:** heterotrophic denitrification, metagenomics, psychrotolerant denitrifiers, nitrate removal, constructed wetland

## Abstract

Denitrifying woodchip bioreactors (WBR), which aim to reduce nitrate (NO_3_^−^) pollution from agricultural drainage water, are less efficient when cold temperatures slow down the microbial transformation processes. Conducting bioaugmentation could potentially increase the NO_3_^−^ removal efficiency during these specific periods. First, it is necessary to investigate denitrifying microbial populations in these facilities and understand their temperature responses. We hypothesized that seasonal changes and subsequent adaptations of microbial populations would allow for enrichment of cold-adapted denitrifying bacterial populations with potential use for bioaugmentation. Woodchip material was sampled from an operating WBR during spring, fall, and winter and used for enrichments of denitrifiers that were characterized by studies of metagenomics and temperature dependence of NO_3_^−^ depletion. The successful enrichment of psychrotolerant denitrifiers was supported by the differences in temperature response, with the apparent domination of the phylum *Proteobacteria* and the genus *Pseudomonas*. The enrichments were found to have different microbiomes’ composition and they mainly differed with native woodchip microbiomes by a lower abundance of the genus *Flavobacterium*. Overall, the performance and composition of the enriched denitrifying population from the WBR microbiome indicated a potential for efficient NO_3_^−^ removal at cold temperatures that could be stimulated by the addition of selected cold-adapted denitrifying bacteria.

## 1. Introduction

Agriculture is the major source of nitrate (NO_3_^−^) losses to natural water bodies, due to the extensive and increasing global use of nitrogen (N) fertilizers in crop production [1,2]. Leaching of NO_3_^−^ from the upper soil horizon to artificial drains facilitates the losses of N to streams, lakes, and estuaries, where the N loading may cause eutrophication and hypoxia [3]. In Denmark, approximately 50% of the agricultural area is artificially drained [4], and localized water and nutrient management strategies are essential to comply with EU directives on the protection of the aquatic environment [5]. Therefore, denitrifying woodchip bioreactors (WBRs) have been introduced as an on-site technology to mitigate agricultural NO_3_^−^ losses [6]. These bioreactors are based on microbial conversion of NO_3_^−^ to atmospheric N gases and are well suited to treat the polluted water coming directly from subsurface drains [7,8]. Nitrate removal may be virtually complete under conditions with high hydraulic residence time and temperature [9], but WBRs typically show annual mean efficiencies of about 50% [10,11,12]. This reflects that the NO_3_^−^ removal efficiency may drop to 10–20% at low water temperatures (~5 °C), due to the temperature response of the woodchip microbiome [11,12,13]. This drop in NO_3_^−^ removal efficiency is a notable drawback in climate zones where the temperature is low during the main agricultural drainage season [14].

Nitrate removal in WBRs is primarily based on heterotrophic bacterial denitrification [8], which is an anaerobic respiration with sequential reduction of NO_3_^−^ and nitrite (NO_2_^−^) to the gaseous N forms of nitric oxide (NO), nitrous oxide (N_2_O), and dinitrogen (N_2_). Each step in denitrification is catalyzed by independent enzymes in a modular process and the energy metabolism, using NO_x_ as the electron acceptor, is coupled to the oxidation of labile carbon (C) compounds [15,16,17]. In WBRs, the labile C compounds are derived from complex woodchip polymers that are degraded by lignocellulolytic microorganisms [18]. The genus *Cellulomonas* was recently found to harbor bacteria with lignocellulolytic and denitrifying properties [19], but generally different microbial groups are involved in fermentative woodchip decomposition and subsequent complete oxidation by denitrification [9]. Hence, the efficiency of NO_3_^−^ removal at cold temperatures could be limited by the supply of labile C by fermentation and/or the kinetics of denitrification [20]. Psychrophilic denitrifying bacteria, with optimal activity and growth at <15 °C, have been isolated from permanently cold environments [21], but the occurrence and role of cold-adapted bacteria in full-scale WBRs is poorly known [19,22]. It is not clear, for example, if the microbial community of WBRs changes in response to seasonal temperatures [9], as seen in natural ecosystems [23,24], and whether this would allow for the enrichment of cold-adapted denitrifiers, which could be cultivated and used for bioaugmentation to increase the NO_3_^−^ removal efficiency [12,25]. Improved understanding of the temperature response of WBR microbiomes is therefore an important next step towards the development of strategies to improve the NO_3_^−^ removal from agricultural drainage water at cold temperatures.

Here, we study the microbial composition and temperature response of denitrifying enrichments from a full-scale WBR sampled at three different seasons (spring, fall, winter). We used an enrichment approach in order to obtain the response of cultivable bacteria that could play a role in potential bioaugmentation strategies. The performance of the enrichments was characterized in terms of NO_3_^−^ depletion, denitrification, and growth at 5, 10, 20, and 30 °C. Further, the enrichments, as well as the native microbiome of the woodchips, were characterized by shotgun metagenomic sequencing to document the taxonomic and functional characteristics of NO_3_^−^ removal.

## 2. Materials and Methods

### 2.1. Woodchip Bioreactor Facility and Sampling

The study was performed with woodchips from an operating WBR at Serupgård in Jutland, Denmark (N 56.376441, E 9.599318). The WBR consists of a 1.4-m deep basin (L × W, 41.2 × 13.2 m) with willow woodchips (2–32 mm diameter; Ny Vraa Bioenergi I/S, Tylstrup, Denmark) and has a sedimentation pond in front of the water inlet. The WBR receives drainage water from a 128-ha catchment area with arable fields grown in a traditional crop rotation with small grain, pea, and oil seed rape with regular input of mineral and organic fertilizer. The drainage season lasts from early fall to late spring with a break in summer, as typical for Danish hydrological conditions [26]. Drain water temperatures follow a seasonal pattern, typically ranging from a minimum of 2 °C in winter to 11 °C in fall. The outlet of the bioreactor drains into a small stream, eventually merging with the river Gudenå.

Woodchips were sampled from the WBR in spring (8 April 2019), fall (16 September 2019) and winter (27 January 2020) at drain water temperatures of 7.3, 10.7, and 5.9 °C, respectively. Three samples were retrieved from 1-m depth at the center of the WBR using a handheld auger (diameter, 10 cm). The woodchips were pooled in sterile plastic bags, mixed, and used for enrichments upon arrival at the laboratory.

### 2.2. Enrichment Cultures

After each sampling, an inoculum was prepared from representative 50-g woodchip samples by shaking in 120 mL of autoclaved pH-neutral M9 buffer overnight at 20 °C (M9 buffer; 8.8 g of Na_2_HPO_4_, 3.0 g of KH_2_PO_4_, 4.0 g of NaCl, and 0.2 g of MgSO_4_ in 1000 mL of deionized water) to extractmicrobes from the woodchips. Enrichment cultures were prepared by inoculating the microbial extractin M9 buffer (12 mL) into anoxic minimal denitrifying medium (MD medium) composed of M9 buffer supplemented with 300 mg L^−1^ of NO_3_^−^ (KNO_3_), 10 mM of acetate (NaCH_3_COO), and traces of yeast extract (50 µg L^−1^) to ensure vitamins and co-factors for microbial growth. The medium was distributed in 200-mL serum bottles with butyl stoppers (and crimp seals), where oxygen (O_2_) was stripped by flushing the medium with helium (He) gas [27]. After autoclaving (121 °C, 15 min), the cooled medium was inoculated (10%, *v*/*v*) and incubated anoxically in the dark at 10 °C. General anoxic procedures were performed using Hungate techniques [28]. The enrichment lines were transferred to fresh medium 7–10 times when the increase in turbidity indicated microbial growth and/or NO_3_^−^ removal was verified using Spectroquant analytical test kits (Merck Millipore, Darmstadt, Germany) based on ISO standards [29].

### 2.3. Temperature Dependence of NO_3_^−^ Depletion, Denitrification and Growth

The temperature response of the final enrichment cultures was tested at 5, 10, 20, and 30 °C. Batch cultures were incubated in MD medium with four biological replicates per temperature. Indices of (i) nitrate reduction, (ii) denitrification, and (iii) bacterial growth were quantified as: (i) depletion of NO_3_^−^, (ii) production of N_2_O in cultures amended with acetylene (C_2_H_2_) to inhibit nitrous oxide reductase [30], and (iii) increase in optical density (OD) measured at 600 nm (OD_600_) [31].

As a general procedure, 10% inoculum with OD_600_ of 0.16–0.17 was transferred to 200-mL serum bottles with 120 mL of anoxic MD medium. The bottles were injected with C_2_H_2_ (10% of the headspace volume) through a butyl stopper and incubated at 5, 10, 20, and 30 °C with hourly temperature logging. Samples of headspace gas (0.2 mL) were withdrawn at daily intervals and transferred to He-filled 6-mL Exetainer vials for measurement of N_2_O by gas chromatography as previously described [32]. Liquid samples (1 mL) were withdrawn for analysis of NO_3_^−^ and ammonium (NH_4_^+^) using Spectroquant test kits and for spectrophotometric OD600 measurement (Ultrospec 3300, Amersham Biosciences, Little Chalfont, UK).

### 2.4. Metagenomic Analyses

Two of the four replicate enrichments at 10 °C from each season were prepared for shotgun metagenomic sequencing. To compare the microbiomes in enrichments and environmental samples, extracts of fresh woodchip microbiomes were included in the analyses in fall (20-g woodchip samples shaken overnight in 50 mL of sterile M9 buffer). All enrichments and extracts were centrifuged to harvest microbial cells (14,000× *g*, 10 min), and the pellets were treated with an enzyme mix to enhance cells lysis (Lysozyme, Sigma Aldrich, St. Louis, MO, USA). DNA was extracted using a QIAGEN DNA kit (Cat No. 51126, QIAGEN, Hilden, Germany) following the instructions of the manufacturer, and a DNA library was prepared with a Hamilton NGS STAR system and KapaHyper Plus kit (ROCHE Diagnostics A/S, Hvidovre, Denmark) using 100 ng DNA diluted in 12.5 µL of 10 mM Tris (pH, 7.5–8.5). Fragmentation time was 5 min and ligation was followed by 0.8× double size selection with NucleoMag beads (Macherey Nagel, Dueren, Germany). Seven PCR cycles were run and a single bead cleanup was used post amplification. The library was quantified on a plate reader using Quantit dsDNA reagent (Thermo Fisher Scientific, Roskilde, Denmark). Metagenomes were sequenced using an Illumina MiSeq system with 600 cycles sequencing kit (v3), and 2 × 300 base pairs (bp) paired end reads with dual TruSeq 8 bp indexes. The sequencing depth was 1.4–1.9 Mbp. The fastq sequences of the metagenomes were uploaded on the server MG-RAST [33]. MG-RAST uses SEED-subsystem database and KEGG database [34] to assign taxonomy and functional attributes to the microbial communities. Taxonomic diversity of sequences reads were interpreted using M5RNA (multi-source non-redundant ribosomal RNA) database with a threshold of >15 bp per contig, an E-value cut-off = 1 × 10^−5^, and a sequence identity of 60% [35] (Appendix A Table A1). Finally, the sequences were annotated using the database RefSeq [36] for taxonomical identification and the Kyoto Encyclopedia of Genes and Genomes (KEGG) Orthology for functional characterization. Data obtained from MG-RAST were normalized to prevent bias due to sequences length difference. The functional analysis aimed at genes associated with the pathway of denitrification (Appendix A Table A2) including nitrate reductases (*nar*, *nap*), nitrite reductases (*nir*), nitric oxide reductases (*nor*), and nitrous oxide reductases (*nosZ*) as well as genes (*nrfA*) associated with dissimilatory nitrate reduction to ammonia (DNRA).

### 2.5. Calculations and Statistical Analyses

The rate of NO_3_^−^ depletion at each temperature was calculated from linear regression of initial time course data, representing up to 95% NO_3_^−^ depletion. The temperature dependence of NO_3_^−^ depletion rates was analyzed by fitting the log transformed Arrhenius equation (Equation (1)) to the data using ordinary least squares linear regression:(1)$$\ln Rate=\ln A-\frac{E_{a}}{RT}$$
where *A* is the frequency factor, *E_a_* is the activation energy (J mol^−1^), *R* is the gas constant (8.31 J K^−1^ mol^−1^), and *T* is the temperature in Kelvin [37]. The temperature coefficient, *Q*_10_, defined as the factor by which the rate increases with a 10 degrees rise in temperature (i.e., from *T* to *T* + 10), was calculated from *E_a_* as:(2)$$Q_{10}=exp\left(\frac{10E_{a}}{RT\left(T+10\right)}\right)$$

Accumulation of N_2_O in the batch cultures was calculated as the sum of gaseous and dissolved N_2_O according to:(3)$$N_{2}O\;\left(mass\right)=\frac{C_{g}\left(V_{g}+V_{l}\times \alpha \right)\times M}{V_{temp,gas}}$$
where *C_g_* is the headspace N_2_O concentration, *V_g_* is the headspace volume, *V_l_* is the liquid volume, *α* is the temperature-specific Bunsen solubility coefficient for N_2_O, *M* is the mole mass of N_2_O, and *V_temp,gas_* is the temperature-specific molar gas volume [38].

Statistical analyses were performed with the software R 3.6.1 [39] to compare the NO_3_^−^ depletion rates at different seasons and incubation temperatures. Mixed linear models were used to describe NO_3_^−^ depletion rates in subsets of the different temperatures as function of the fixed parameter ‘sampling season’ and the random parameter ‘individual sample’, using the R package nlme [40]. Post-hoc pairwise comparisons of rates (*n* = 4) at the same incubation temperature, but representing different seasons, were made at a significance level of *p* < 0.05 using the method Tukey.

K-means clustering (k neighbors = 4, repeat = 100) was performed with the package *ComplexHeatmap* [41] to cluster the genomic samples based on the taxonomic abundance of the 45 most abundant species.

## 3. Results

### 3.1. Nitrate Depletion, Denitrification and Growth

Enrichments from all seasons showed full NO_3_^−^ depletion (>95%) within 2–3 d at 20 and 30 °C (Figure 1). However, at 10 °C, only winter enrichments showed full NO_3_^−^ depletion within 2–3 d, as compared with 7 d for spring and fall enrichments. Winter enrichments also showed complete NO_3_^−^ depletion at 5 °C, whereas spring and fall enrichments showed only 30–60% NO_3_^−^ removal within 7 d (Figure 1). Rates of NO_3_^−^ depletion accordingly were highest for winter enrichments at all individual temperatures (Appendix A Table A3).

Concurrent N_2_O accumulation confirmed that denitrification occurred and that the process at 5–10 °C was faster in the winter enrichment as compared with spring and fall enrichments (Figure 1). However, the final N_2_O accumulation at 10 °C was highest in the enrichment spring. The apparent ratio of NO_3_^−^-N depletion to N_2_O-N accumulation indicated >40% conversion at 10–30 °C and <40% conversion at 5 °C. However, at 5 °C, there was an apparent time lag between NO_3_^−^ depletion and N_2_O formation, likely due to the transformation rate of intermediates.

Nitrate depletion and N_2_O accumulation were accompanied by microbial growth at all temperatures (Figure 1). At 5 and 10 °C, growth was faster and reached a higher biomass for enrichment cultures from winter than from spring and fall.

The temperature dependence of NO_3_^−^ removal rates was fitted well by the Arrhenius equation for the enrichment from winter (*R*^2^ = 0.93), with slightly weaker fits for spring (*R*^2^ = 0.87) and fall (*R*^2^ = 0.85), mostly due to over-prediction at 30 °C (Figure 2). Activation energies (*E_a_*) ranged from 47–65 kJ mol^−1^ across the enrichments, with resulting *Q*_10_ of 2.0 for the winter enrichment and 2.5–2.7 for the spring and fall enrichments, as determined for the temperature interval from 5–15 °C (Figure 2). These temperature coefficients indicated that the denitrifying activity in the winter enrichment was less sensitive to temperature decreases than the spring and fall enrichments.

Collectively, the results of the temperature assays of NO_3_^−^ depletion, denitrification and growth indicated that the microbial enrichments from the winter season had more efficient NO_3_^−^ removal at 5–10 °C than the enrichments from spring and fall.

### 3.2. Microbiome Composition and Functional Genes

The microbial communities of the enrichments were dominated by the phylum *Proteobacteria* with high representation of the orders *Pseudomonadales*, *Burkholderiales*, *Neisseriales,* and *Enterobacteriales* (Appendix A Figure A1). Heatmaps and clustering showed a consistent microbiome similarity between enrichments from each sampling season as compared with the enrichments from other seasons and native woodchip microbiomes (Figure 3). The genus *Pseudomonas* had the highest relative abundance (9–45%) in most enrichments (Figure 3), thus showing a competitive ability to proliferate at the enrichment temperature of 10 °C. As an exception, one replicate of winter enrichments was characterized by the notable abundance of the genus *Arthrobacter* (Figure 3). The metagenomes obtained from woodchip samples showed a high abundance of the order *Flavobacteriales* and genus *Flavobacterium*, which was not found in the enrichment cultures (Figure 3).

The full suite of denitrification genes was found in all enrichments and native woodchip microbiomes generally with the predominance of the NO_3_^−^ reductases *narG* and *napA* (Figure 4). Ratios of nitrate reductases to nitrite reductases, (*nar* + *nap*)/*nir*, were in the range 6–12 for the enrichments from spring and winter, and around 1.5–2 for native microbiomes and the enrichments from fall. The ratio of *nir*/*nosZ* was 0.6–0.7 in native WBR microbiomes and close to 1.0 in the enrichments. The *nrfA* genes associated with DNRA were found in minor proportions but seemed relatively more abundant in the enrichments from spring (Figure 4).

## 4. Discussion

### 4.1. Temperature Dependence of Nitrate Removal

Enrichment cultures from spring, fall, and winter showed a psychrotolerant temperature response with NO_3_^−^ consumption and growth at 5 °C but optimum between 20 and 30 °C [42,43]. Nitrate depletion and growth proceeded without lag phases in the temperature assays (with inoculum from exponential growth at 10 °C), showing that the bacteria enriched at 10 °C were active at the alternate temperatures. Substantial production of N_2_O documented denitrification as a major pathway of NO_3_^−^ depletion, but an exact mass balance could not be established, as N uptake in biomass was not measured and also it was uncertain if the acetylene assay was completely efficient in blocking N_2_O reduction to N_2_ [44]. Furthermore, lag phases were observed for N_2_O accumulation, notably at 5–10 °C for the spring and fall enrichments, indicating the slow transformation of denitrification intermediates, such as NO_2_^−^ and NO (or potentially intracellular NO_3_^−^), although these pools were not quantified in this study.

The temperature dependence of NO_3_^−^ depletion was characterized by *E_a_* of 47–65 kJ mol^−1^ (*Q*_10_ of 2.0–2.7), which corresponded to temperature coefficients of denitrification found in other ecosystems [45] and meta-anlyses of denitrifying bioreactors [46]. Winter enrichments had the lowest *Q*_10_, indicating the most robust response to temperature decreases. This is consistent with an adaptation of the WBR microbiome to environmental temperature changes throughout the year, although the present results from enrichment cultures may not quantitatively describe such effects. Seasonal changes in the denitrifying community of a saltmarsh sediment were demonstrated by King and Nedwell [23], who found that a distinct psychrotrophic population developed to a maximum size at the end of the winter but disappeared during summer. This was attributed to the seasonal selection of different denitrifiers, and enrichment studies suggested that the psychrotrophs, which became apparent during winter, belonged to the genus *Pseudomonas* [23].

The efficient NO_3_^−^ removal by winter enrichments at low temperature could be considered surprising, since low NO_3_^−^ removal efficiencies (~20%) may be observed at operating Danish WBR facilities during winter [11]. However, this contrast between enrichment and field observations essentially illustrates that the WBR microbiome holds a potential for efficient NO_3_^−^ removal at cold temperatures, but this potential may need to be stimulated under field conditions for better WBR performance [14]. Since C limitation was alleviated by acetate in the present study, we measured a direct temperature response of the denitrifying population, which was more robust for the winter enrichment. Under field conditions, however, the temperature response of NO_3_^−^ removal may be controlled also by the temperature response of lignocellulolytic woodchip degraders, which provide the simpler organic substrates that are completely oxidized by denitrifiers [47]. Bioaugmentation to optimize the NO_3_^−^ removal efficiency of WBRs may therefore have to concurrently exploit cold-adapted denitrifiers and lignocellulolytic microorganisms, but so far further studies are needed to clarify the interaction between fermenters and denitrifiers controlling the NO_3_^−^ removal rates in WBRs [9,20,48].

### 4.2. Taxonomic and Functional Composition

Bacterial species from the most abundant orders in the enrichments have previously been identified as dissimilatory NO_3_^−^ reducers [49,50,51,52]. Notably, the high relative abundance of *Pseudomonadales* aligned with the ecology of pseudomonads as opportunistic heterotrophs in nutrient-rich environments [53], often being numerically dominant denitrifiers in soil ecosystems and WBRs [19,22,54,55]. Indeed, since operating WBRs may experience periods of shifting NO_3_^−^ availability and redox conditions (e.g., due to breaks in drainage water flow), these ecosystems may favor versatile microbial ecophysiologies like pseudomonads, which are able to switch between anaerobic and aerobic respiration, due to O_2_-controlled expression of denitrification genes [56,57].

Winter enrichments, which showed efficient NO_3_^−^ removal at cold temperatures, also had *Burkholderiales* as a dominant order with *Acidovorax* (Figure 3) and *Polaromonas* (data not shown) as abundant genera. Bacteria from the order *Burkholderiales* have previously been identified as key denitrifiers in WBRs [9,52] and, e.g., *Acidovorax* has also been found as active denitrifiers in activated sludge [58] and wetland ecosystems [59]. Moreover, *Polaromonas* harbors several psychrotolerant denitrifying species [60,61] and was also identified as abundant denitrifiers in WBR facilities in Minnesota (USA) operated at a water temperature of 15 °C [19]. *Burkholderiales* therefore appears to be an important component of cold-adapted microbiomes in WBRs.

The native microbial populations in the woodchips resembled the enrichments in terms of high relative abundance of the genus *Pseudomonas*. The main difference was related to the abundance of the genus *Flavobacterium*, which was not obtained in the enrichment cultures. Species of *Flavobacterium* are generally aerobic heterotrophs, many of which are psychrotolerant [62], but the genus also harbors aerobic and facultative anaerobic denitrifiers, thus demonstrating metabolic versatility [63,64,65]. The genus was also found in relatively high abundance in previous studies of denitrifiers from wastewater treatment facilities [66] and bioreactors [19,47]. The absence of *Flavobacterium* in our denitrifying enrichment cultures indicated the limitations of the enrichment approach, which was performed under anoxic conditions but apparently introduced selective pressures hampering competitive proliferation of *Flavobacterium*.

Nitrate reductases showed the highest relative abundance among the denitrification genes, notably for the enrichment cultures from spring and winter. One reason could be that *nar* genes are present in many microorganisms that can reduce NO_3_^−^ to NO_2_^−^, but lack further denitrification genes [67,68]. *Nar* genes are present also in DNRA bacteria [69], but the low relative abundance of *nrfA* genes showed that the DNRA pathway of NO_3_^−^ conversion was minor in both enrichments and native WBR microbiomes. Furthermore, despite the slighlty increased *nrfA* abundance in some enrichments (spring), this did not result in increased NH_4_^+^ concentrations, which always remained below 5 mg N L^−1^ (data not shown). Ratios of *nir*/*nosZ* indicated similar genetic potential for production and consumption of N_2_O in the enrichment cultures and even an overcapacity of N_2_O consumption in the native woodchip microbiome, which could be related to the presence of non-denitrifying N_2_O reducers, found to play an important ecological role in many terrestrial ecosystems [70]. However, although gene numbers reflect a genetic potential, it is not a direct measure of realized metabolic activity [71]. Thus, under environmental conditions, the N_2_O/N_2_ product ratio in bacteria with genetic capacity for full denitrification is controlled by factors such as pH and availability C and N, which may fluctuate over time in WBRs [20]. Resulting differences in N_2_O emissions within and between different full-scale WBRs were empirically documented in a recent Danish study, but the differences were not clearly reflected by the gene distributions (J. Audet, personal communication).

The observations at 5 °C of NO_3_^−^ removal with a lag phase for N_2_O production (Figure 1) might indicate a temperature dependent expression of denitrification genes or a different temperature sensitivity of different denitrification steps, where uptake and reduction of NO_3_^−^ to NO_2_^−^ could be favored at cold temperatures. During a study on denitrification genes abundance in a constructed wetland, Chon et al. [72] also observed higher *nar* abundance in winter (~2 °C) than in spring, which they attributed to seasonal shift in microbial populations or at least in metabolic activity. Similarly, Liao et al. [73] observed a high relative abundance of *nar* at low water temperature (10 °C), which concurred with incomplete denitrification and NO_2_^−^ accumulation in the effluent of a lab-scale expanded granular sludge bed reactor. In environmental settings, however, NO_2_^−^ is rapidly turned over by abiotic and biotic processes [74] and the ecological consequence of possible intermediate NO_2_^−^ accumulation at cold temperatures in WBRs remains to be documented. In relation aquatic recipients, annual studies of N oxyanions in different Danish WBRs showed NO_2_^−^ concentrations consistently below detection limit in the outlet water (<0.001 mg N L^−1^) thus suggesting negligible environmental impact (J. Audet, personal communication).

### 4.3. Conclusions and Perspectives

Bacterial NO_3_^−^ reducing enrichments from an operating full-scale WBR were obtained, which showed a psychrotolerant temperature response, with growth and activity at 5 °C, notably for cultures started with samples collected in winter. These cultures also showed the most robust response to temperature decreases with a *Q*_10_ of 2.0. *Pseudomonas* was generally the dominating bacterial genus in enrichments and native woodchip microbiomes, and denitrification was the dominating pathway of dissimilatory NO_3_^−^ reduction as indicated by functional gene analyses. In their entirety, the results indicated that even though low NO_3_^−^ removal efficiencies may occur in field-scale WBRs during winter [11], the microbiomes harbor denitrifying bacteria adapted to cold temperatures, which are cultivable and show efficient NO_3_^−^ reduction, e.g., at 5 °C. This conclusion corroborates results from laboratory denitrifying bioreactors [14] and other denitrification beds [75,76]. Enumeration of denitrifying bacteria was not performed in the present study, but results from wetland ecosystems have shown a dynamic fluctuation of different temperature groups of denitrifies in response to seasonal temperature changes [23]. Strategies to improve the NO_3_^−^ removal from agricultural drainage water at cold temperatures, may comprise the cultivation of psychrotolerant denitrifiers, as shown in the present study, and using these microorganisms for bioaugmentation in WBRs to establish higher population densities than naturally arise by seasonal dynamics, thereby further increasing the enzymatic potential for NO_3_^−^ transformation. This will necessitate the selection of strains with competitive ability to establish over time in the WRBs [12]. At the same time, however, it should be established to what extent denitrification at cold temperatures in full-scale WBRs is co-limited by the temperature response of lignocellulolytic woodchip degraders, which provide the electron and C sources for heterotrophic denitrification.

## Figures and Tables

**Figure 1 microorganisms-09-01331-f001:**
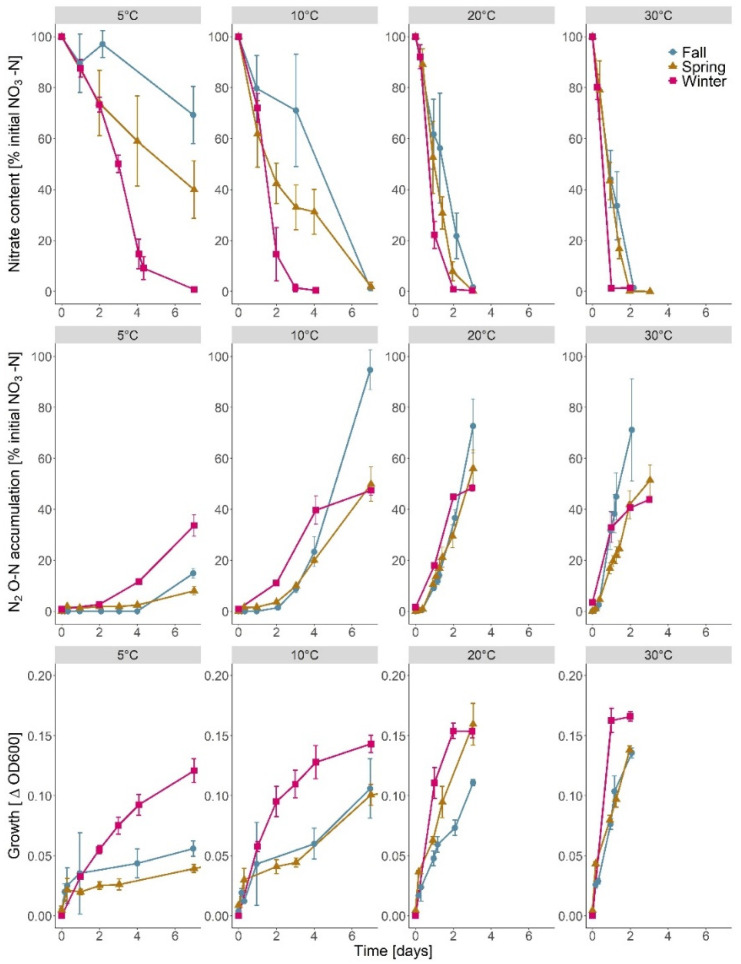
Time course of nitrate (NO_3_^−^) depletion (first row), denitrification (second row), and growth (third row) of NO_3_^−^ reducing enrichments from a woodchip bioreactor sampled in spring fall, and winter. Enrichments were assayed at 5, 10, 20, and 30 °C. Denitrification was assayed as N_2_O production in cultures amended with acetylene, and growth was measured as the increase in optical density at 600 nm (∆OD_600_).

**Figure 2 microorganisms-09-01331-f002:**
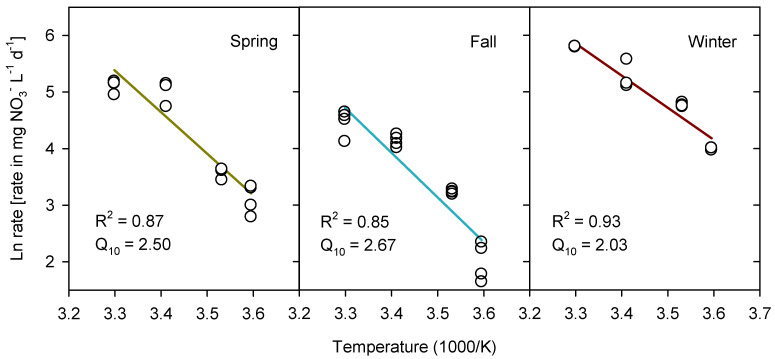
Arrhenius plots of the temperature effect on nitrate (NO_3_^−^) removal rates in enrichments from woodchip bioreactor samples from three different seasons (spring, fall and winter). Lines show linear regression between data points (circles). Coefficients of determination (*R*^2^) and temperature coefficients (*Q*_10_), calculated from Arrhenius activation energies, are shown.

**Figure 3 microorganisms-09-01331-f003:**
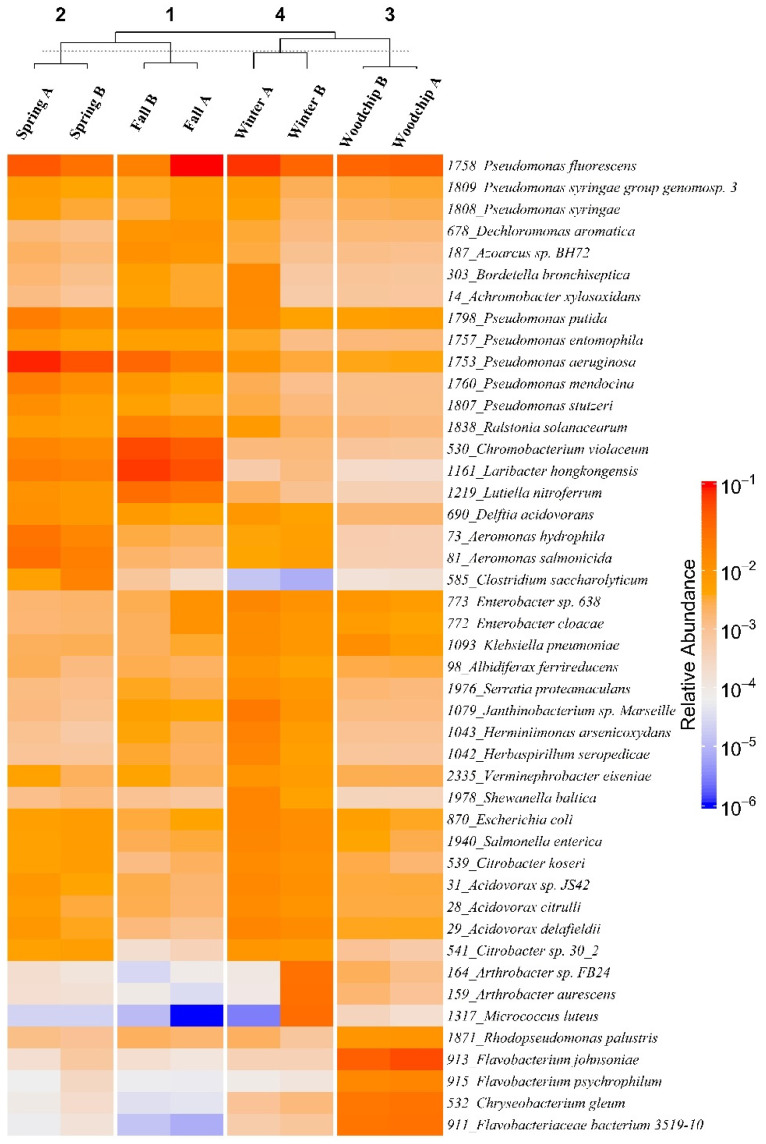
Heatmap and cluster analysis of enrichments (spring, fall, and winter) and native woodchip samples (fall) based on the 45 relatively most abundant bacterial species. The A and B notations refer to biological replicates. Each row of the heatmap displays the abundance of an Operational Taxonomic Unit (OTU) for which the corresponding number and the identified species are displayed on the right-hand side. The number on top of the figures represents the different clusters identified with k-means clustering (k = 4).

**Figure 4 microorganisms-09-01331-f004:**
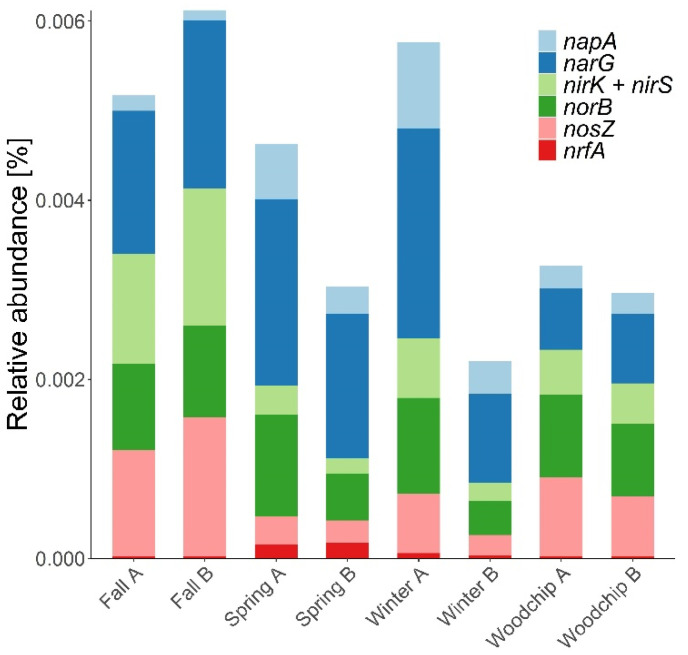
Relative abundance of genes related to denitrification and dissimilatory nitrate reduction to ammonia (*nrfA*) in enrichments (spring, fall and winter) and native woodchip samples (fall). Denitrification genes included nitrate reductases (*nar*, *nap*), nitrite reductases (*nir*), nitric oxide reductases (*nor*) and nitrous oxide reductases (*nosZ*). The results are presented as the number of hits attributed to each gene in percent of the total number of hits obtained from the samples with the KO database [35]. The A and B notations refer to biological replicates.

## Data Availability

The datasets presented in this study can be found in online repositories. The names of the repository/repositories and accession number(s) can be found below: ENA/PRJRB45276. Additional information is available in Table A1.

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
