# Peer review of "Temperature Sensitivity and Composition of Nitrate-Reducing Microbiomes from a Full-Scale Woodchip Bioreactor Treating Agricultural Drainage Water"

_microorganisms, 2021, doi:10.3390/microorganisms9061331_

Round 1

Reviewer 1 Report

microorganisms-1256735

The authors of this study enriched phychrotolerant nitrate-reducing bacteria by using the woodchip samples collected from a full-scale woodchip bioreactor in three different seasons (spring, fall, and winter) and analyzed their nitrate removal and N2O production in different temperatures. They also characterized the enrichment cultures by metagenomics. They found that the enrichment cultures started with samples collected in winter had the highest nitrate removal at low temperatures.

General comments

Low temperatures are known to limit the N removal performance of woodchip bioreactors. This study is trying to address this important issue. The manuscript is generally well written. The experiments are carefully designed and performed. Results were interesting and well interpreted/discussed.

Specific comments

L81: N 56.376441, E 9.599318

L97: Please show the volume of the M9 medium used in this step.

L98: Why samples were kept in M9 medium at 20C overnight? What is the purpose of this step? To release microbes from woodchips?

L100: This sentence is a little confusing because the seed microbes were not enrichment cultures. Maybe “Microbes released from the woodchip samples were grown …” or “Enrichment cultures were prepared by inoculating the M9 buffer (X ml) into anoxic MD medium …”?

L128: Why only 10C samples were used for metagenomic analysis? 5C samples (or comparison of cultures enriched under different temperatures) would be more interesting to analyze.

Fig. 3: How did you obtain the OTU? What genes did you use for OTU clustering? What was the cutoff? Please describe these in the Method section.

Fig. 4: How did you calculate the relative abundance? The number of reads identified as the target gene divided by the total number of reads? If you compare the relative abundances of multiple genes, I think it is important to normalize the abundance by gene length because longer genes have more chance to be mapped than shorter genes.

L236: Why nap was not included in the calculation [(nar+nap)/nir instead of nar/nir]?

Table 1: Length --> Yield  

Author Response

Answer to specific comments:

L81: Updated L81

L98: The purpose of this step was to recover microbes from the woodchips to start an enrichment. This step was not designed to be selective but merely to recover all kind of microbes from the woodchips. The mild temperature coupled with a long time provided favorable conditions for all type of microbes to transfer to the medium. This specification has been added to the main text (L101).

L100: The suggestion of the reviewer for more clarity is very welcomed. The text has been modified to answer this comment (L101-102).

L128: The different generations of enrichment cultures were incubated at 10℃. Therefore, to be able to compare with the main enrichment culture lines it was decided to conduct the metagenomics on the samples incubated at 10℃. We agree that it would be very interesting to investigate the composition at 5℃ but we had to make choices for the analyses (L130).

Fig.3: MG-RAST uses SEED-subsystem database together with KEGG database to assign taxonomy and functional attributes to the microbial communities. The taxonomic diversity of sequences reads were interpreted using M5RNA (multisource non-redundant ribosomal RNA) database on MG-RAST server with an E-value cut-off of 1×10-5 and sequence identity of 60%. The Method has been updated to provide more specific information (L146-151).

Fig. 4: MG-RAST is providing a “hit” abundance. This “hit” refers to a match between a sequence of amino acids in the sample and a given annotation. We based our study on these “hits”. The relative abundance is the number of hits/matches found for a given annotation divided by the total amount of hits/matches for all the annotations. By using this method, we compare the abundance of the specific genes without needing to normalize for the length of each genes.

L236: We initially chose to communicate only the ratio of nar/nir to harmonize our study with some existing literature. However, the comment of the reviewer made us realize that it would be more relevant to include the ratio with both nitrate reductases (nar+nap)/nir (L243-244).

Table 1: This column of Table 1 communicates base pair counts conducted by MG-RAST prior to the metagenomics analysis pipeline. We have updated the column names to make it clearer to the reader.

Reviewer 2 Report

In this study, Jéglot and colleagues enriched for denitrifying microorganisms from woodchip bioreactors sampled in the fall, winter and spring. Their goal was to determine if bioaugmentation with a carbon source could boost denitrification rates in WRBs in cooler climates. Using these enriched communities, denitrification rates over various temperatures were monitored in addition analysis of metagenome sequence data from enrichment cultures and woodchip samples from the fall. Overall, the enrichment community from the winter samples outperformed the spring and fall enrichments on the basis of total nitrate reduction at lower temperatures and the rate of nitrate reduction at all temperatures tested. Predominant genes involved in denitrification, and predominant taxa were noted however not with respect to one another.

Comments

  • More detail on how the metagenome sequence data was analyzed needs to be included in the Methods section. For example, on what basis were OTUs defined?
  • Section 4.2 discusses the composition of the enrichment communities and predicts their functional role without taxonomic analysis of the contigs that denitrification genes were identified from. For example, in lines 293-294 the authors propose pseudomonads as the primary microbes catalyzing denitrification and in lines 301-302 they also note that the order Burkholderiales were noted as primary denitrifiers in different studies. Further analysis of the shotgun metagenomics data from this study should help to identify which members are encoding for the denitrification genes quantified in Figure 4.
  • Line 56 and elsewhere: lignocellulolytic is misspelled.

Author Response

Answer to comments:

More information relative to OTUs definition and MG-RAST analysis pipeline was added to the Method section (L146-151).

Section 4.2: The reviewer raises a relevant point. MG-RAST does not provide the phylogenetic classification of the genes and the authors of the study decided to keep the metagenomics analysis based on MG-RAST output. However, this potential improvement of the metagenomics analysis has been taken into consideration and will surely be addressed in future studies.

Misspells of lignocellulolytic have been corrected.